

# Miocene Antarctic ice sheet area responds significantly faster than volume to CO$_2$-induced climate change

Lennert B. Stap[1], Constantijn J. Berends[1], and Roderik S.W. van de Wal[1,2]

[1]Institute for Marine and Atmospheric research Utrecht, Utrecht University, 3584 CC Utrecht, the Netherlands
[2]Faculty of Geosciences, Department of Physical Geography, Utrecht University, Utrecht, the Netherlands

**Correspondence:** L.B. Stap (L.B.Stap@uu.nl)

**Abstract.** The strongly varying benthic $\delta^{18}$O levels of the early and mid-Miocene (23 to 14 Myr ago) are primarily caused by a combination of changes in Antarctic ice sheet (AIS) volume and deep ocean temperatures. These factors are coupled since AIS changes affect deep ocean temperatures. It has recently been argued that this is due to changes in ice sheet area rather than volume, because area changes affect the surface albedo. This would be important when the transient AIS grows relatively
faster in extent than in thickness, which we test here. We analyse simulations of Miocene AIS variability carried out using the three-dimensional ice-sheet model IMAU-ICE forced by warm (high CO$_2$, no ice) and cold (low CO$_2$, large East-AIS) climate snapshots. These simulations comprise equilibrium and idealised quasi-orbital transient runs with strongly varying CO$_2$ levels (280 to 840 ppm). Our simulations show limited direct effect of East-AIS changes on Miocene orbital timescale benthic $\delta^{18}$O variability, because of the slow build-up of volume. However, we find that AIS area responds significantly faster and more
strongly than volume to the applied forcing variability. Consequently, during certain intervals the ice sheet is receding at the margins, while ice is still building up in the interior. That means the AIS does not adapt to a changing equilibrium size at the same rate or with the same sign everywhere. Our results indicate that the Miocene Antarctic ice sheet affects deep ocean temperatures more than its volume suggests.

## 1   Introduction

Orbital scale variability of benthic $\delta^{18}$O levels during the early and mid-Miocene (23 to 14 Myr ago) primarily reflects changes in Antarctic Ice Sheet (AIS) volume and deep ocean temperatures. The separate contributions of these factors to the benthic $\delta^{18}$O signal is fiercely debated. On one side of the argument, geological studies argue for strong AIS dynamism (Pekar and DeConto, 2006; Shevenell et al., 2008), with ice periodically advancing and retreating e.g. over Wilkes Land (Sangiorgi et al., 2018) and the Ross Sea sector (Hauptvogel and Passchier, 2012; Levy et al., 2016; Pérez et al., 2021). Oppositely, studies
based on ice-proximal sea-surface temperature records from dinoflagellate cyst assemblages (Bijl et al., 2018) and TEX$_{86}$ data (Hartman et al., 2018) call for limited AIS variability. Ice sheet modelling as deployed here can be used to reconcile AIS variability with the forcing climate evolution.





Ultimately, AIS volume and deep ocean temperatures changes are determined by insolation, regulated by the prevailing
climatic background, e.g., geography (Stärz et al., 2017; Colleoni et al., 2018; Paxman et al., 2020; Halberstadt et al., 2021),
vegetation (Knorr et al., 2011), and greenhouse gas concentrations (Frigola et al., 2021; Burls et al., 2021; Gasson et al., 2016;
Halberstadt et al., 2021). AIS volume and ocean temperatures are in fact also directly related, because the AIS size affects sea
water temperatures, doing so in different ways at different ocean depths. Climate model studies have found a spatially hetero-
geneous, but overall warming of Southern Ocean sea surface temperatures due to increasing AIS volume during the Middle
Miocene Climatic Transition (MMCT; ~14 million years ago), through the local wind field (Knorr and Lohmann, 2014) or
through a decreased upwelling longwave flux over the elevated ice sheet (Frigola et al., 2021). These models also show a
negligible (Knorr and Lohmann, 2014) or cooling (Frigola et al., 2021) response of deep ocean temperatures to an increasing
AIS size. The contrasting responses between these studies can be ascribed to a difference in the implementation of ice sheet
size changes. The former study (Knorr and Lohmann, 2014) only included the effect of ice thickness differences, keeping the
ice area constant, whereas the latter (Frigola et al., 2021) also studied different AIS areal extents. A recent study confirmed that
indeed ice area, rather than ice volume, affects deep ocean temperatures, because ice area impacts on the all-important surface
albedo (Bradshaw et al., 2021).

This has ramifications for the partitioning between the contributions of AIS volume and deep ocean temperatures changes
to benthic $\delta^{18}$O levels during the early and mid-Miocene. The ice area may adapt significantly faster to climatic changes than
ice volume, in other words the AIS may grow more quickly in extent than in thickness. The effect of ice sheet changes on deep
ocean temperatures can then vary comparatively more strongly than the ice sheet volume suggests. Whether this holds from
an ice-physical perspective needs to be tested, preferably through transient AIS simulations in which the timescale of ice sheet
adjustment is implicitly captured.

Here, we analyse simulations of Miocene AIS variability (Stap et al., 2022) carried out using the three-dimensional ice-sheet
model IMAU-ICE (De Boer et al., 2014; Berends et al., 2018) forced by warm (high $CO_2$, no ice) and cold (low $CO_2$, large
East-AIS) climate snapshots generated by the general circulation model GENESIS (Burls et al., 2021). The AIS simulations
comprise equilibrium and idealised quasi-orbital transient runs (40 to 400 kyr timescales) with strongly varying $CO_2$ levels
(280 to 840 ppm). Utilising a recently developed matrix interpolation method (Berends et al., 2018), the climate forcing is
interpolated based on $CO_2$ levels as well as varying ice sheet configurations, capturing key ice-sheet-atmosphere feedbacks.
Stap et al. (2022) investigated the effect of including these feedbacks in the interpolation of climate forcing and found that
their net effect is to reduce simulated transient Miocene Antarctic ice-sheet variability. Here, we focus on the relation between
AIS volume and area and its effect on AIS growth and decay on orbital timescales. Our simulations corroborate the thesis put
forth by Bradshaw et al. (2021) that the pre-MMCT AIS area responds significantly faster than volume to $CO_2$-induced climate
change.



## 2 Models and methods

For this research, we analyse simulations of the AIS that were priorly conducted using the three-dimensional ice-sheet model IMAU-ICE v1.1.1-MIO (De Boer et al., 2014; Berends et al., 2018; Stap et al., 2021). The model set-up and simulations are
described in detail in Stap et al. (2022), here we include a brief summary. IMAU-ICE combines the shallow ice approximation (SIA) and shallow shelf approximation (SSA) to simulate the Antarctic ice sheet and shelf dynamics, on a 40-km square grid. The groundling line is not treated in any special manner. The ice-free Antarctic bedrock topography and bathymetry are taken from geological reconstructions of the early Miocene (24 to 23 Myr ago) (Paxman et al., 2019; Hochmuth et al., 2020a).

The basal mass balance underneath the floating ice shelves is parameterised (Pollard and DeConto, 2009; De Boer et al., 2013), in part based on a linear relation to $CO_2$-dependent ocean temperature change (Beckmann and Goosse, 2003; Martin et al., 2011; De Boer et al., 2013), while calving is not included. Though the ocean forcing has limited impact on our results (Stap et al., 2022), the treatment of the mass balance of floating ice and of the grounding line in our current modelling effort are in need of improvement (see Sect. 4). These are focal points of ongoing model development (Berends et al., 2022a, b).

The surface mass balance is calculated using a simple but effective insolation-temperature-melt model (Bintanja et al., 2002; Fettweis et al., 2020; Stap et al., 2022) with monthly forcing precipitation and surface air temperature input from pre-run Miocene climate simulations by the general circulation model GENESIS v3.0 (Thompson and Pollard, 1997; DeConto et al., 2012; Burls et al., 2021). Using the Miocene global paleotopography from Herold et al. (2008) in combination with specific
Antarctic topography obtained from ice-sheet simulations by DeConto and Pollard (2003), and constant present-day insolation, GENESIS simulates global climate on a T31 spectral resolution grid. Dynamic sea ice and vegetation models ($2° \times 2°$ resolution) are connected to the atmospheric model, as well as a slab-ocean component (50-m resolution). Multiple GENESIS simulations were carried out (Burls et al., 2021), of which we use a cold simulation forced with a $CO_2$ level of 280 ppm and a large East-Antarctic ice sheet, and a warm simulation with 840 ppm $CO_2$ and no ice (Stap et al., 2022). A matrix interpolation
method (Berends et al., 2018; Stap et al., 2022) is deployed to interpolate between warm and cold simulations in order to obtain the transient forcing precipitation and surface air temperature fields applied at any time. The interpolation is based on a combination of the (external) $CO_2$ forcing, and the simulated ice sheet, so that essential long-term ice-sheet-atmosphere interactions are taken into account. These interactions affect both accumulation and ablation. They include the albedo-temperature feedback, the surface-height-mass-balance feedback, and the ice-sheet desertification effect, i.e., depletion of precipitation over
ice sheets crescending in size (e.g. Oerlemans, 2004).

In this study, we investigate the reference experiment presented by Stap et al. (2022), a simulation set consisting of eleven equilibrium and three transient simulations (Stap et al., 2021). The equilibrium simulations were conducted keeping the $CO_2$ concentration stable at various levels for 150 thousand model years, starting either with no ice (ascending branch) or with a
developed ice sheet (return branch). In addition to the existing simulations, six extra equilibrium simulations were conducted



for the current study at critical $CO_2$ levels where large ice volume changes take place: at 420, 448, 476 pm in the ascending branch, and at 644, 672 and 700 ppm in the return branch. In the transient simulations, the $CO_2$ level was lowered from 840 to 280 ppm in a linear fashion, and thereafter gradually raised back to 840 ppm. This V-pattern $CO_2$ forcing was executed in 100 kyr, 400 kyr, and sequentially five times in 200 kyr (40 kyr per cycle) (Fig. S1). We here analyse ice volume and ice area output generated every 1000 model years.

## 3 Results

In the transient simulations, the evolving ice sheet area and volume can be considered to be adjusting towards continuously changing equilibrium states (Stap et al., 2020). We distinguish three stages in the growth phase of the ice sheet (Fig. 1A-C, Table 1). Initially, the $CO_2$ level is lowered, and the ice sheet area and volume both grow, trailing behind the equilibrium ice sheet size (Fig. 2). Therefore, the difference between the (ascending branch) equilibrium volume and the transient volume, henceforth the volume deficit, increases. The volume growth rate increases approximately proportionally to this volume deficit during this stage (Fig. 3; pink and lightblue lines). In the subsequent second growth stage, after about 40% of the simulation time (40 kyr for the 100-kyr simulation, and 159 kyr for the 400-kyr simulation) the volume deficit starts to decrease. This is when the $CO_2$ level drops below 392 ppm and the increase of the equilibrium volume levels off. The transient ice area and volume are both still smaller than equilibrium and continue to increase (Fig. 2). Because the ice area is now larger than before, the ice volume accumulates over a wider area, causing the ice volume growth rates to be larger with respect to the ice volume deficit (Fig. 3; red and blue lines). After 52 kyr in the 100-kyr simulation and 201 kyr in the 400-kyr simulation, 52% and 50% of the total run time respectively, the transient ice area attains its maximum. We use a 10-kyr moving average to determine this turning point (Fig. S1B), because due to discretization flickering area variations occur when ice shelves are formed. In the ensuing third growth stage, the ice sheet area declines but because the ice thickness still increases in the interior, so does the total ice volume (Fig. 2). In contrast to growth stages I and II when volume growth was driven by area growth, the AIS now has a warming effect on deep ocean temperatures despite the growing volume. During growth stage III, the ice volume growth rate quickly drops until the growth phase ends after 66 kyr (66% of the run) in the 100-kyr simulation and after 228 kyr (57% of the run) in the 400-kyr simulation (Fig. 3; brown and darkblue lines).

In both transient simulations, a similar maximum area is attained, meaning the ice sheet area is controlled more by the $CO_2$ level than by the ice-sheet-climate interaction. The effect of the AIS on deep ocean temperatures would therefore also be similar. The smaller volume-to-area ratio in the 100-kyr simulation compared to the 400-kyr simulation (Fig. S2) means that the mean ice thickness and hence the surface height are lower. The ice sheet desertification effect is therefore reduced and hence precipitation rates are larger. As a result, the volume growth rates are much larger in the shorter 100-kyr simulation than in the 400-kyr simulation (Fig. 4A).





Whereas the ice sheet growth rates vary rather smoothly, waning of the ice sheet occurs in self-sustained bursts (Fig. 4B). This is due to instabilities caused by ice-sheet-climate interaction: initial ice sheet decay triggers further climate warming through surface lowering and darkening, which causes further decay, etcetera. Successive decay bursts in the 400-kyr simulation are related to ice sheet collapse in Coats Land, Princess Elizabeth Land, Wilkes Land and George V Land, Dome Fuji, Dome Circe and finally Dome Argus. In the 100-kyr simulation, the collapse of Princess Elizabeth Land, Wilkes Land and George V Land, and of Dome Fuji and Dome Circe happen simultaneously (Figs. 1D and 5). Hence, peak decay rates are larger than in the 400-kyr simulation (Fig. 4B). Because ice sheet decay is more erratic than ice sheet growth, the relation between volume excess (the transient ice volume minus the (return branch) equilibrium volume) and the volume decay rate (Fig. 6) is less straightforward than the relation between volume deficit and volume growth rates (Fig. 3). In general, our simulated Miocene AIS diminishes much faster than it grows, which causes a sawtooth shape of ice sheet variability similar to Pleistocene glacial cycles (Stap et al., 2019, 2022).

In the 400-kyr simulation the $CO_2$ level is altered more gradually than in the 100-kyr simulation. The ice sheet remains closer to equilibrium as it has more time to adapt (Stap et al., 2022). It is furthermore clear that in our simulations the ice sheet area responds faster and more strongly to $CO_2$-induced climate change than the volume. These differences in phase and relative amplitude (compared to the equilibrium size) between the responses of ice volume and area are also larger in the 100-kyr simulation than in the 400-kyr simulation, and they are magnified further still in our 40-kyr simulation (Figs. 2 and S1).

## 4   Discussion

AIS volume and deep ocean temperature changes constitute orbital timescale variations of benthic $\delta^{18}O$ levels during the early and mid-Miocene. These factors are determined by climate change, which in our simulations is induced by strongly varying $CO_2$ levels (280 to 840 ppm). Earlier ice-sheet modelling studies have shown that the relatively dry Antarctic climate, even during ice-free times, leads to a slow build-up of the AIS (Stap et al., 2019, 2022). This limits the variability of AIS on orbital timescales, causing a reduced and delayed contribution to the benthic $\delta^{18}O$ signal (Fig. 7, green and purple lines). Through their direct effect on the climate and hence deep ocean temperatures, slow AIS changes would also imprint on the contribution of deep ocean temperatures to the benthic $\delta^{18}O$ signal. However, recent climate model studies have revealed that ice area, rather than ice volume, is decisive for this influence (Knorr and Lohmann, 2014; Frigola et al., 2021; Bradshaw et al., 2021), adding another factor to be considered. Here, we have shown that ice area responds significantly faster to climate change than ice volume. We deduce this from the earlier occurence and larger magnitude of the transient ice area maximum compared to the ice volume maximum, relative to their maximum sizes as obtained from the 280-ppm equilibrium simulation (Fig. 7, cyan and purple lines). Consequently, the Antarctic ice sheet affects deep ocean temperatures generally more strongly than what can be inferred from ice volume variability. Arguably, this is partly due to the specifics of the matrix interpolation scheme for the climate forcing that we use. However, simulations using a simpler index method, in which the interpolation is solely based on the $CO_2$ level, show qualitatively similar results (Fig. S3). Our results provide the potential for a shorter lag between





deep ocean temperature and climate (here $CO_2$) variability. In reality, this lag will depend first of all on how much deep ocean temperatures are affected by the AIS. Bradshaw et al. (2021) showed a simulated ∼2 K decrease in deep ocean temperature caused by a drop in $CO_2$ from 850 to 280 ppm, and a further ∼1 K added by the emergence of a continent-covering AIS. The climate-deep ocean lag will also depend on the timescale of deep ocean temperature adjustment, which we cannot constrain

using our model set-up and needs to be investigated further. A shorter lag between deep ocean temperatures and climate would cause a further tipping of the balance towards a stronger contribution to Miocene benthic $\delta^{18}O$ variability by deep ocean temperatures compared to the contribution by ice volume changes.

In earlier research, the transiently evolving Miocene AIS was analysed in terms of adjustment towards changing equilibrium

states (Stap et al., 2020). This analysis was performed using a model based on the notion that an ice sheet will grow when it is smaller and shrink when it is larger, than its explicitly prescribed equilibrium size. We have shown here, however, that it is not sufficient to regard the AIS as a whole in this respect. During our transient simulations, there are intervals when the margins of the ice sheet are receding while ice is still being built up in the interior. This discrepancy in phasing between area and volume affects the relation between the volume growth rate and volume deficit in the growth phase. Moreover, the decay

phase is characterised by large self-sustained bursts of ice mass loss ignited by ice-sheet-climate feedbacks. In Sect. S4, we show that in our model similar behavior is exhibited by the North American ice sheet in settings representative for Pleistocene glacial-interglacial variability (Scherrenberg et al., 2023). This finding disputes conceptual models of glacial variability that (implicitly) assume identical timescales for ice area and ice volume adjustment.

Our model set-up will be extended in future research with some important factors that are still missing now. For instance, insolation changes, the ultimate driver of orbital variability, are not considered here. As a first step, they can be integrated in the matrix interpolation method (e.g. Ladant et al., 2014; Tan et al., 2018), but ultimately a carbon cycle model should be included in the model set-up to facilitate using insolation changes as the only external forcing (Ganopolski and Brovkin, 2017). Additionally, noisier forcing variability compared to our idealised smooth quasi-orbital forcing may reduce the mean size

of the ice sheet (Niu et al., 2019; Stap et al., 2020). Furthermore, our simulations are aggravated by poorly represented ocean forcing (see Sect. 2). Ocean forcing was demonstrated to have limited impact on the transiently evolving AIS in the simulations investigated here, since AIS changes are mainly constricted to the East side of Antarctica (Stap et al., 2022). However, recent data (Marschalek et al., 2021) and modelling results (Gasson et al., 2016; Halberstadt et al., 2021) have made a case for a dynamic West AIS during the Miocene. Modelling the generally lower-lying West AIS requires an accurate incorporation of

ocean forcing. This is currently being improved in IMAU-ICE (Berends et al., 2022b), but outside the scope of our present effort. A related model improvement that will be deployed in future investigations of Miocene AIS variability concerns the grounding line physics (Berends et al., 2022a). While this may increase the speed of ice sheet retreat, the main findings of our current study, which primarily concern the growth phase, will likely not be affected.

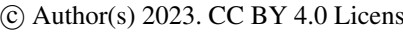

## 5    Conclusions

We have analysed equilibrium and idealised transient simulations of the response of the Miocene AIS to strong $CO_2$ variations
(280 to 840 ppm). Our simulations support the premise that transient Miocene AIS area growth significantly outpaces ice vol-
ume growth (Bradshaw et al., 2021). Consequently, the relative variability of ice area is larger compared to ice volume. Ice
area controls the effect of the AIS on deep ocean temperatures through surface albedo (Bradshaw et al., 2021). We therefore
conclude that on orbital timescales during the early and mid-Miocene (pre-MMCT) the AIS affects deep ocean temperatures
more than its volume suggests.

The discrepancy between ice area and ice volume growth means that the transiently changing ice sheet does not adjust
towards equilibrium at the same rate or even with the same sign everywhere. Three stages can be distinguished during ice sheet
growth. Initially, the ice area and mean thickness increase, trailing behind the growth of the equilibrium size. During the second
stage, the equilibrium size levels off, but the transient ice area and thickness continue to increase. During the third stage, the ice
sheet is receding at the margins, while ice is still building up in the interior. While during stages I and II the AIS has a cooling
effect on deep ocean temperatures, this turns into a warming effect during stage III as the ice area declines. Finally, the decay
of the AIS occurs in self-sustained bursts due to instabilities caused by ice-sheet-climate interaction.

*Code availability.* The code for IMAU-ICE v1.1.1-MIO is available from https://github.com/IMAU-paleo/IMAU-ICE/releases/tag/v1.1.1-
MIO (last access: 10 March 2023) and https://doi.org/10.5281/zenodo.6352125 (Berends and Stap, 2021).

*Data availability.* The data analysed in this study are openly accessible from the PANGAEA database (Stap et al., 2021).

*Author contributions.* LBS designed the research and performed the experiments, with technical assistance from CJB. LBS, CJB, and
RSWvdW analysed the results. LBS drafted the paper, with input from all co-authors.

*Competing interests.* The authors declare that they have no conflict of interest.

*Acknowledgements.* L.B. Stap is funded by the Dutch Research Council (NWO), through VENI grant VI.Veni.202.031. Simulations were
performed on the Gemini computing cluster of the Faculty of Science, Utrecht University. We thank Edward Gasson for providing the
GENESIS climate input data for our simulations, and Meike Scherrenberg for providing the model set-up for the Pleistocene North American
ice-sheet simulations discussed in the Supplement. We further thank Gregor Knorr for commenting on an earlier draft of the manuscript.



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





**Table 1.** Indication of the change in mean ice thickness, ice area, volume deficit (equilibrium volume minus transient volume), and a description of the various stages during ice sheet growth.

| Stage | Thickness | Area | Deficit | Description |
|-------|-----------|------|---------|-------------|
| I | Increases | Increases | Increases | Initial growth trails equilibrium |
| II | Increases | Increases | Declines | Equilibrium volume levels off |
| III | Increases | Declines | Declines | Thickness increase outweighs declining area |





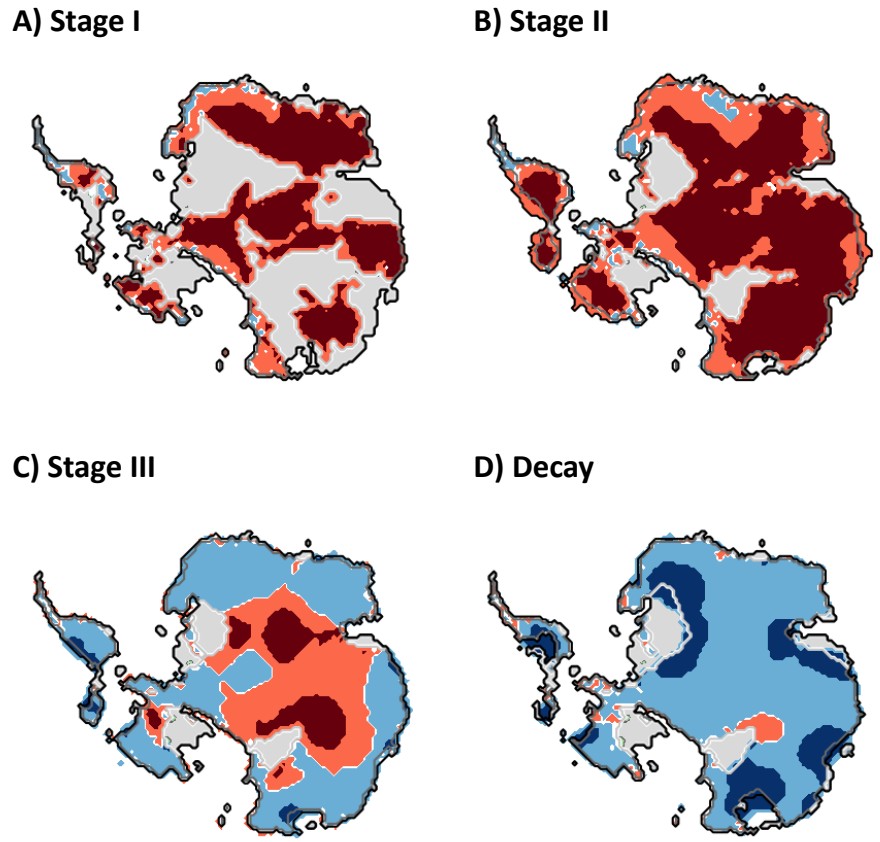

**Figure 1.** Change in Antarctic ice sheet thickness in the 100-kyr simulation, between **(A)** 30 kyr and 40 kyr (growth phase Stage I), **(B)** 40 kyr and 50 kyr (Stage II), **(C)** 60 kyr and 70 kyr (Stage III), and **(D)** 70 kyr and 80 kyr (decay phase). Red colors indicate ice thickness increase, blue colors decrease. Dark colors indicate where the ice thickness change exceeds 100 m over 10 kyr. Grey areas indicate ice-free land and white areas ocean.



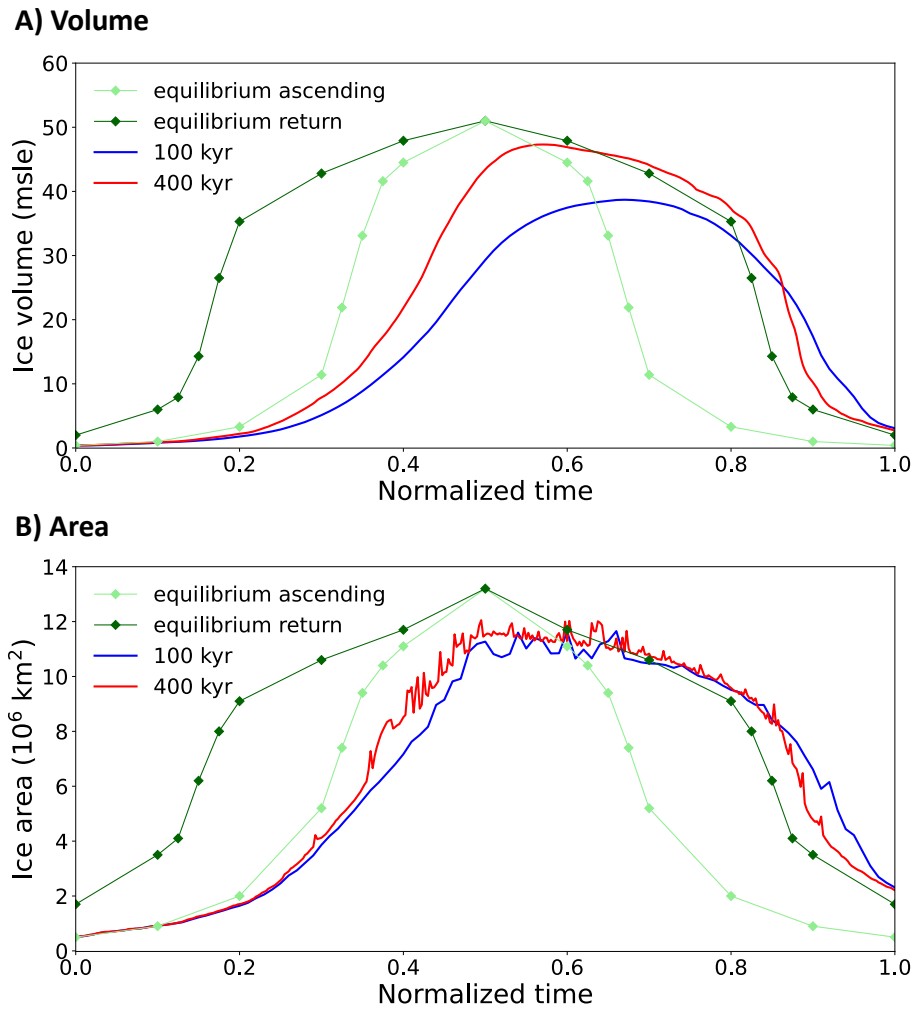

**Figure 2. (A)** Transient evolution of ice volume over time, normalized with respect to the maximum integration time, for the 100-kyr (blue) and 400-kyr (red) simulations. **(B)** Same for ice area. The connected symbols indicate the ascending branch (lightgreen) and return branch (darkgreen) equilibrium ice volume and area pertaining to the prevailing $CO_2$ level.





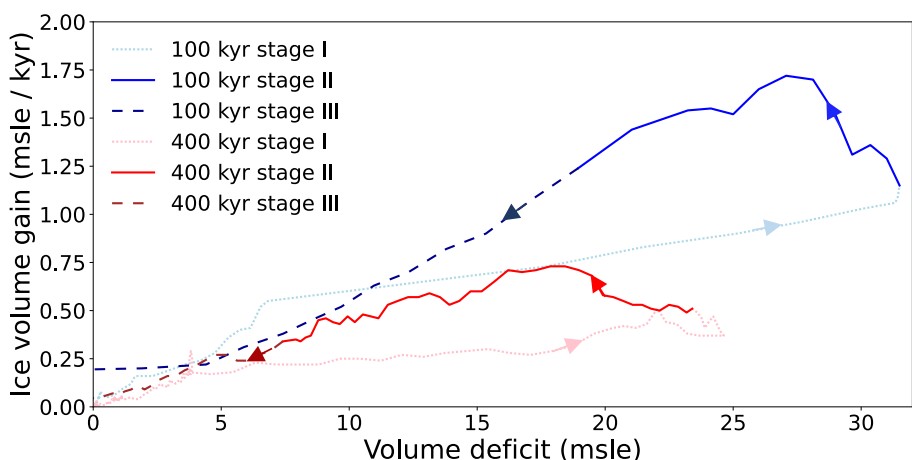

**Figure 3.** Growth rate plotted against the volume deficit, i.e., the difference between the (ascending branch) equilibrium volume and the transient volume, for the 100-kyr (blue) and 400-kyr (red) simulations. The different stages (I, II, and III) explained in the main text and Table 1, are indicated by varying color shades and dashes. The arrows indicate the progression direction.

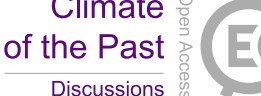

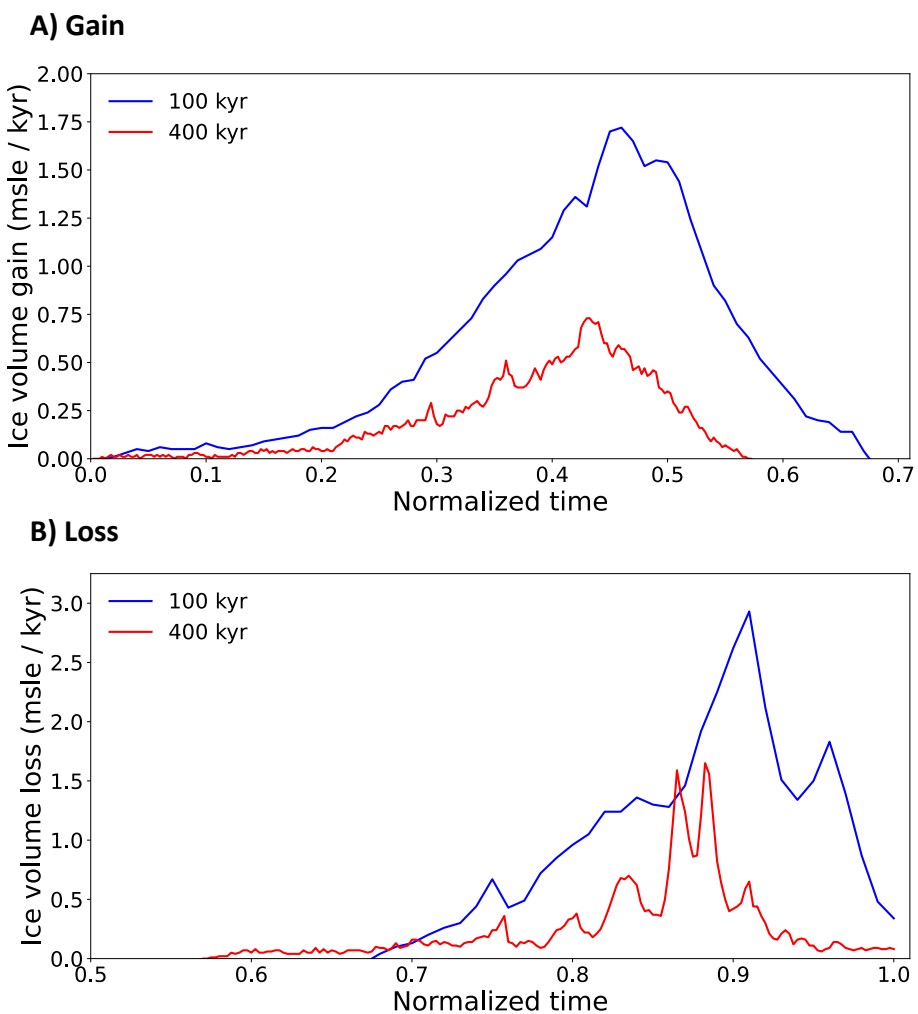

**Figure 4. (a)** Volume growth rate over normalized time, and **(b)** decay rate over time, for the 100-kyr (blue) and 400-kyr (red) simulations. Mind the differing axis scales.

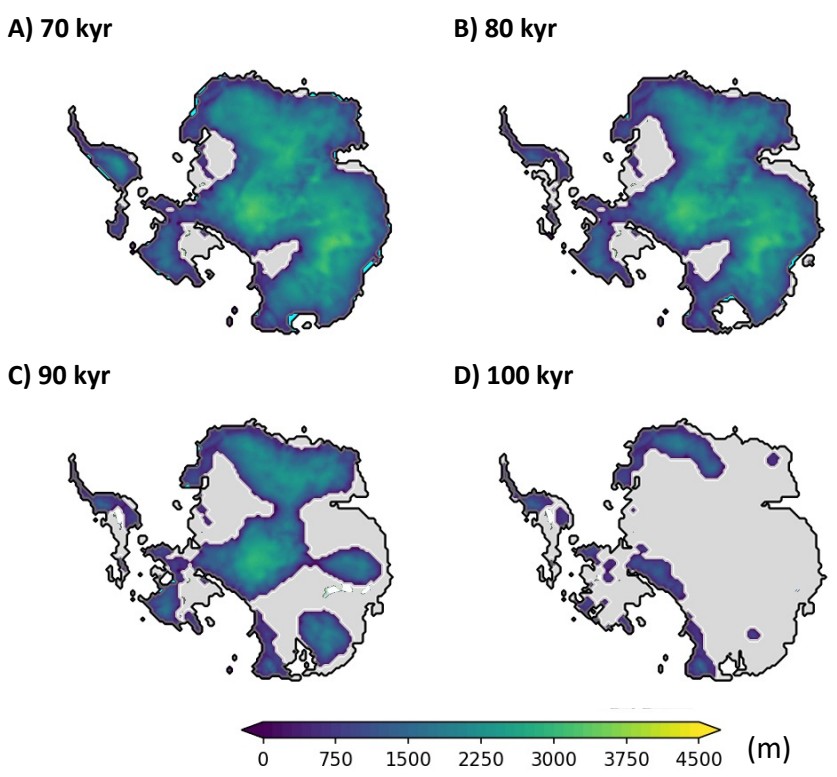

**Figure 5.** Maps of the ice thickness during the decay phase in the 100-kyr simulation, after **(A)** 70 kyr, **(B)** 80 kyr, **(C)** 90 kyr, and **(D)** 100 kyr. Grey areas indicate ice-free land and white areas ocean.



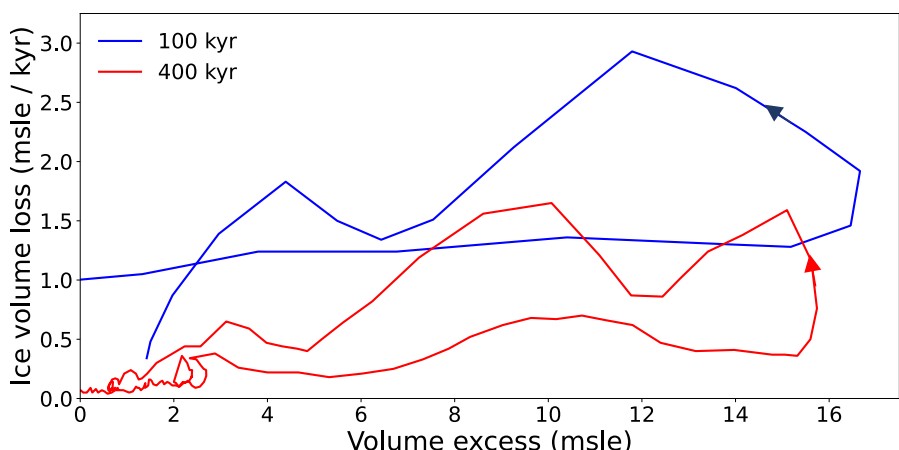

**Figure 6.** Volume decay rate plotted against the volume excess, i.e., the difference between the transient volume and the (return branch) equilibrium volume, for the 100-kyr (blue) and 400-kyr (red) simulations. The arrows indicate the progression direction.





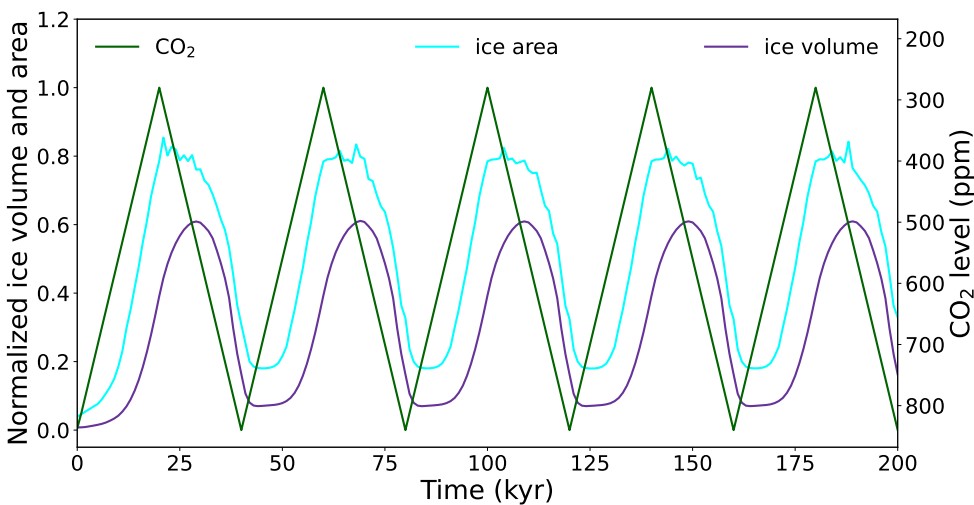

**Figure 7.** Transient evolution over time of ice area (cyan) and ice volume (purple) relative to their maximum sizes as obtained from the 280-ppm equilibrium simulation, $13.2 \times 10^6$ km$^2$ and 51.0 msle respectively, for the 40-kyr simulation. The green line shows the forcing $CO_2$ level. The right y-axis is reversed because $CO_2$ is generally negatively related to the benthic $\delta^{18}O$ signal.