# Peer review of "Miocene Antarctic ice sheet area adapts significantly faster than volume to CO2-induced climate change"

_Climate of the Past, 2023_

## Author Response (AR1)

**AUTHOR'S RESPONSE**

Reviewer comments in regular font (page/line numbers refer to initial submission)
**Author responses in bold font**
*Changes made to the manuscript in bold italics (page/line numbers refer to the track-changed manuscript).*

**Reply to Referee #1**

This paper presents a series of numerical experiments of Miocene Antarctic ice sheet(AIS) variability using IMAU-ICE, forced by idealized climate forcing. They focus relative `speed' (I will come back for this term) of growth rate of ice-covered area and ice volume, and conclude that the area responds faster and stronger than the volume. Consequently, it is possible to have shrink in the area and growth in the volume at the same time, which may indicate that the Miocene Antarctic ice sheet affects deep ocean temperatures more than its reconstructed volume suggests. I think this paper is fairly well written with some exception below, and can be accepted with major revision.

**We would like to thank the reviewer for a careful examination of our work. We are pleased they are generally positive about our study. Below, we address their comments.**

The first point is small, but may be essential. I agree that the main conclusion that the area responds `faster' than the volume, however, it requires the definition of `fast' in this context. It may impress that the authors compare the time derivative of ice volume and that of the area, which is not comparable. I understand the authors do not mean in this sense. It is clear that they compare the relative speed to approach to the equilibrium sizes of the two aspects. L41 in introduction has already such sentences: The ice area adapt..... This is good, and better to emphasize again in the result section.

**We agree that an absolute comparison between area and volume growth is fallacious. We will make it more clear in the revised manuscript that we compare them only in a relative sense, in terms of adjustment towards changing equilibrium states. We will choose the more accurate word 'adapt' over 'respond' for the title and throughout the revised manuscript.**

*Changes: title, abstract (lines 9-10), page 2 line 41, page 5 lines 145-147, page 6 line 160, and page 7 line 216.*

The second point is also small. It is true, as the authors said, that they analyze both equilibrium and transient experiments (e.g., the first sentence of the conclusion), but their relative weights are much different. The results of equilibrium experiments are merely used as the reference of the transient experiments, and few information about the equilibrium experiments are presented. This is just because they already presented enough analysis in their previous paper. I suppose it is better to clearly mention that the detail analysis of equilibrium experiments are already done, and that they are used as the reference of transient experiment in the present paper.

**Indeed, the focus of the paper is on the transient experiments. The equilibrium experiments serve as a reference, albeit an important reference considering the previous comment of the reviewer and our response. We will clearly state this in the introduction and method sections of the revised manuscript.**

*Changes: page 2 lines 49-50, and page 3 lines 94-95.*

About ice-sheet climate feedback. I cited two blocks where it is mentioned:

L124: This is due to instabilities caused by ice-sheet-climate interaction: initial ice sheet decay triggers further climate warming through surface lowering and darkening, which causes further decay, etcetera.

L169: Moreover, the decay phase is characterised by large self-sustained bursts of ice mass loss ignited by ice-sheet-climate feedbacks.

It should be true, but need more information to discuss whether this is correct. There are several possibilities to have such rapid variation in ice sheet dynamics. For example, some instabilities are known for marine ice sheet. However, this is not the case in the present paper, since marine ice-sheet is much limited. For another example, there is a thermodynamic instability (binge-purge theory) in ice-sheet system. Also, this may not be the case because thickness is less small to have high temperature near the base. If such discussion is inserted, the confidence for ice-climate interaction will be much higher.

**We will include a discussion on the cause of the decay bursts in the discussion section of the revised manuscript. To aid the discussion, we will compare our reference experiment to additional experiments that exclude the albedo-temperature feedback, and that use LGM-like sub-shelf melt rates (these simulations were already introduced in our previous work: Stap et al., 2022). We will keep the discussion short though since we would like to keep our main focus on the growth phase.**

*Changes: pages 6-7 lines 189-197, and included Fig. S6*

Finally, the explanation of the experiment used in the Fig. 7 is not sufficient. Actually this is another experiment, but it may be not suitable to describe in the experiment section because it is just additional for further analysis. At least its objective and the configuration should be described in the discussion section.

**The 40-kyr simulation most clearly demonstrates that transient ice area variability remains closer to equilibrium with the CO2 forcing than ice volume as it covers multiple relatively short-term forcing cycles. We agree that the very brief introduction of this simulation in the method section is insufficient. We will therefore more clearly describe the configuration and aim of this simulation in the method and discussion sections. A table describing the simulations we analyse will also be included in the revised manuscript.**

*Changes: page 4 lines 101-102, page 6 lines 161-163, and new Table 1.*

Some other more minor points.

Subfigure labels are mixed. Better to use either small or capital(?) letters in the caption, labels, and also on the context. For example in Figure 4: In the figure and main text A B are used while a b are used in the caption, etc.

**We will uniformly use capital letters for the main text and figure captions in the revised manuscript.**

*Changes: Fig. 4*

L125 and after. There are many location names in the text. I agree that many readers are familiar with them, nevertheless, I still feel to need the map of Antarctica in the present paper to show where they are.

**We will include a map of Antarctica indicating (roughly) the locations mentioned in the text as a supplemental figure.**

*Changes: new Fig. S3*

Table 1. Include `Decay' phase to complete.

**We will include the decay phase in the table.**

*Changes: Table 2*

Figure 1. It is not clear whether the figure have ice-shelf parts. I suppose that the white area is ice-free ocean, and the surrounding black line is either grounding line or coastline. Better to explain them.

**In difference plots such as these, it is hard to indicate ice shelves, as parts may be ice shelf in one simulation and grounded in the other. We therefore chose not to make a distinction between sheet and shelf areas. We will state this in the caption, and explain that the grey grounding lines, and black coastlines pertain to the latest timestep constituting the difference.**

*Changes: caption of Fig. 1*

Figure 2. The explanation of equilibrium ascending and return is a little bit lacked. I suppose adding the corresponding CO2 levels on the horizontal axis may help.

**We will include the CO2 levels in a separate panel.**

*Changes: Fig. 2 (specifically new panel A).*

Figure 4. Adding e.g. light-gray horizontal lines for each 0.5 or 1 msle/kyr may help to mind the different scales.

**We will use the same axis scales in both panels for better comparison.**

*Changes: Fig. 4.*

Figure 7.  Replacing CO2 line by corresponding equilibrium size (area and volume), as figure 2, may be better.  CO2 levels line should be separated.

**The purpose of this figure is to show that the maximum (normalized) ice area is higher that volume and follows the minimum CO2 level more quickly. This is in our opinion most clearly visible when these three variables are plotted together in the same panel.**

**Reply to Referee #2**

Stap et al., investigate the uncertainty surrounding the relative contribution of changes in Antarctic ice sheet volume and deep ocean temperature to the orbital scale variability in Miocene benthic delta18O records. The authors use the IMAU-ICE ice sheet model to conduct equilibrium and idealized transient quasi-orbital simulations of Antarctic ice sheet variability under CO2 concentrations that vary from 280ppm to 840ppm. These simulations are forced by warm (high CO2, no ice) and cold (low CO2, large East Antarctic ice sheet) climate model snapshots. They conclude that the slow buildup of volume changes seen in their simulations indicate that the East Antarctic ice sheet had a limited direct effect on benthic delta18O on orbital timescales during the Miocene. Changes in the area of the Antarctic ice sheet respond faster and affect deep ocean temperatures more than volume changes.

Overall the findings are interesting and merit publication. I have the following relatively minor comments.

**We would like to thank the reviewer for a careful examination of our work. We are pleased they appreciate our study. Below, we address their comments.**

Some suggestions on how to improve the clarity of the manuscript:

- Abstract: "It has recently been argued that this is due to changes in the ice sheet area rather than volume, because area changes affect the surface albedo. This would be important when the transient AIS grows relatively faster in extent than thickness, which we test here" Explicitly state what "this" is.

  **Changed to 'this finding'.**

  *Changes: abstract line 4.*

- A schematic to accompany the text at the beginning of Section 3 and Table 1 would be very helpful.

  **In our opinion, Fig. 1 already offers such a schematic. In combination with the table and main text, this should be enough guidance for the reader to the different stages of ice sheet growth (and decay).**

- Ln 112 – please explain how it has a warming effect, what is the logic or previous study used to infer this?

  **At this stage (stage III), the volume increases but the ice area actually already decreases. Bradshaw et al. (2021) found that the ice area determines the effect on deep ocean temperatures by altering the surface albedo. Hence, the overall effect of the ice sheet change on deep ocean temperatures would be warming. We will describe this mechanism and include a reference to Bradshaw et al. at this point in the revised manuscript.**

  *Changes: page 4 lines 119-121*

- "generated by the general circulation model GENESIS (Burls et al., 2021)." & "Multiple GENESIS simulations were carried out (Burls et al., 2021)," Note that as reported in Table 3 of Burls et al., 2021, I think that the Genesis simulations used were first published by Gasson et al. (2016) Gasson, E., DeConto, R. M., Pollard, D., & Levy, R. H. (2016). Dynamic Antarctic ice sheet during the early to mid-Miocene. Proceedings of the National Academy of Sciences of the United States of America, 113(13), 3459–3464. https://doi.org/10.1073/pnas.1516130113

  **These are in fact different simulations, which use the same set-up save for the coupling to the ice-sheet model. The Antarctic ice sheet is prescribed in the GENESIS simulations that we use here, in contrast to Gasson et al (2016). We will indicate this in the revised manuscript.**

  *Changes: page 3 lines 80-82*

- "Multiple GENESIS simulations were carried out (Burls et al., 2021), of which we use a cold simulation forced with a $CO_2$ level of 280 ppm and a large East-Antarctic ice sheet, and a warm simulation with 840 ppm $CO_2$ and no ice (Stap et al., 2022)." which orbital configuration was used ?

  **Both simulations use present-day orbital settings.**

  *Changes: page 4 line 84*

- "In this study, we investigate the reference experiment presented by Stap et al. (2022), a simulation set consisting of eleven equilibrium and three transient simulations (Stap et al., 2021)." this part will be hard to follow for someone not familiar with the Stap et al. papers, do they use the forcing from the GENESIS simulations too? A table summarizing the simulations used/performed with the summary details for each would help make the methods section clearer.

  **Yes, these simulations use the settings described in the preceding paragraphs. We will point this out in the revised manuscript and include a table of the simulations we analyse.**

  *Changes: page 4 lines 91-92, and new Table 1.*

---

## Author Response (AR2)

**Reply to Referee #1**

Now I am very happy to see that all the concerns I raised at the previous review are solved, except for a minor technical correction appended.

Around L184:
which also excludes marine instabilities as the main cause of the bursts. `marine instabilities' are not a good word. Please reword them, such as `instabilities due to ice-ocean interaction' (better), or `marine ice-sheet instabilities'. Actually, I do not recommend the latter words, because they are ambiguous especially in this context.

**We thank the reviewer for considering our paper once more. We have implemented their remaining suggestion. Furthermore, we have made all relevant model output openly accessible, and updated the data availability statement accordingly.**